# High Spatial-Temporal Resolution Estimation of Ground-Based Global Navigation Satellite System Interferometric Reflectometry (GNSS-IR) Soil Moisture Using the Genetic Algorithm Back Propagation (GA-BP) Neural Network

**Yajie Shi** [1,2], **Chao Ren** [1,2,*] , **Zhiheng Yan** [1,2] and **Jianmin Lai** [1,2]

1   School of Surveying, Mapping and Geographic Information, Guilin University of Technology, Guilin 541004, China; shiyajie6320@glut.edu.cn (Y.S.); 2120201715@glut.edu.cn (Z.Y.); 2120201680@glut.edu.cn (J.L.)
2   Guangxi Key Laboratory of Spatial Information and Surveying and Mapping, Guilin 541004, China
*   Correspondence: renchao@glut.edu.cn

**Abstract:** Soil moisture is one of the critical variables in maintaining the global water cycle balance. Moreover, it plays a vital role in climate change, crop growth, and environmental disaster event monitoring, and it is important to monitor soil moisture continuously. Recently, Global Navigation Satellite System interferometric reflectometry (GNSS-IR) technology has become essential for monitoring soil moisture. However, the sparse distribution of GNSS-IR soil moisture sites has hindered the application of soil moisture products. In this paper, we propose a multi-data fusion soil moisture inversion algorithm based on machine learning. The method uses the Genetic Algorithm Back-Propagation (GA-BP) neural network model, by combining GNSS-IR site data with other surface environmental parameters around the site. In turn, soil moisture is obtained by inversion, and we finally obtain a soil moisture product with a high spatial and temporal resolution of 500 m per day. The multi-surface environmental data include latitude and longitude information, rainfall, air temperature, land cover type, normalized difference vegetation index (NDVI), and four topographic factors (elevation, slope, slope direction, and shading). To maximize the spatial and temporal resolution of the GNSS-IR technique within a machine learning framework, we obtained satisfactory results with a cross-validated R-value of 0.8660 and an ubRMSE of 0.0354. This indicates that the machine learning approach learns the complex nonlinear relationships between soil moisture and the input multi-surface environmental data. The soil moisture products were analyzed compared to the contemporaneous rainfall and National Aeronautics and Space Administration (NASA)'s soil moisture products. The results show that the spatial distribution of the GA-BP inversion soil moisture products is more consistent with rainfall and NASA products, which verifies the feasibility of using this experimental model to generate 500 m per day the GA-BP inversion soil moisture products.

**Keywords:** soil moisture; GNSS-IR technology; high spatial and temporal resolution; GA-BP neural network

## 1. Introduction

Soil moisture is an essential parameter of the global surface water cycle and is also a physical surface quantity that has long been studied with interest. Monitoring soil moisture on a large scale is significant for agriculture, hydrology, and the geographic environment [1]. It also plays a vital role in the climate system and extreme weather such as droughts, floods, and inundation. The persistence of extreme weather is relatively short-lived, and soil moisture has a high memory compared to the atmosphere. In seasonal time scales, soil moisture is of great use [2]. For this reason, it is of more practical importance to achieve soil moisture inversion at a large scale, with high accuracy, low cost, and high spatial and temporal resolution.

The advent of remote sensing technology has made it possible to estimate and monitor surface parameters significantly. Remote sensing satellites can measure soil moisture with uniform accuracy over large spatial scales by constant revisit intervals. Commonly used remote sensing techniques include optical remote sensing and microwave remote sensing. Among the remote sensing techniques capable of measuring soil moisture, microwave remote sensing is the most promising technique for measuring soil moisture with shorter revisit times by sensing the dielectric properties of soil moisture [3,4]. Soil Moisture and Ocean Salinity (SMOS), launched by the European Space Agency, and Soil Moisture Active and Passive (SMAP), launched by NASA in 2015, are the predominant soil moisture missions today [5,6]. Both enable soil moisture monitoring globally by carrying L-band instruments to detect soil moisture in the top 5 cm. It provides soil moisture inversion with a spatial resolution of about 40 km and a revisit time of 2–3 days, with an accuracy requirement of 0.04 $cm^3cm^{-3}$ [7]. Meanwhile, the SMAP mission provides a high spatial and temporal resolution product with 3 km spatial resolution and 2–3 days revisit time with the help of rotating antennas until the hardware fails in mid-2015 [8,9].

However, microwave wavelengths are hundreds to millions of times longer than visible and infrared light, resulting in a low spatial resolution of microwave soil moisture products, which cannot represent localized soil moisture variations in detail. For this reason, a large number of downscaling studies based on the SMOS and SMAP missions have been carried out. It has become an ongoing research hotspot to improve the spatial and temporal resolution of soil moisture products through algorithms. Piles et al. used a combination of a relatively noisy 3 km radar backscatter coefficient and a more accurate 36 km radiometer based on the SMAP task, generating an optimal 10 km soil moisture product with better performance than the reflectance radiometer alon0065 [10]. On this basis, Piles combined the accuracy of SMOS observations with the high spatial resolution of visible/infrared satellite data, effectively capturing soil moisture variability at spatial scales of 10 and 1 km without a significant reduction in root mean square error [11]. With the SMAP mission, Narendra et al. combined coarse-scale radiometry with radar observations detectable at fine-scale spatial heterogeneity; to produce a high-resolution best soil moisture estimate at 9 km, further improving the spatial resolution and accuracy of soil moisture [12]. Knipper et al. even combined SMOS and SMAP missions with information from a high spatial resolution imaging spectrometer to obtain higher resolution (1 km) soil moisture estimates [13]. However, microwave reflections from the soil surface are affected by the state of soil moisture and by environmental factors such as surface roughness, vegetation elements, and interactions with the atmosphere. For this reason, soil moisture inversion using microwave remote sensing is susceptible to non-soil moisture factors and thus error, which may lead to inaccurate soil moisture inversion [14,15]. Therefore, better methods are needed to obtain soil moisture products with high spatial and temporal resolution.

The advent of Global Navigation Satellite Systems (GNSS) has provided us with a new paradigm for monitoring long-time series soil moisture information. It uses the same L-band remote sensing technology as the microwave. The difference, however, is that this technique interferes with the direct signal emitted by GNSS with the reflected signal reflected by the ground at the ground receiver. The interference contains changes caused by differences in the ground surface, which monitors the physical parameters of the Earth's surface [16–19]. In addition, Global Navigation Satellite System-Reflection (GNSS-R) technology and Global Navigation Satellite System-Interferometry (GNSS-IR) technology have been gradually developed based on this technology [20,21]. Because of its advantages, such as all-weather, all-day, and high spatial and temporal resolution, it has been widely used in the fields of soil moisture, sea surface wind field, sea tide, snow depth, and vegetation change [22–27]. For soil moisture research, it can be further divided into ground-based GNSS-IR, airborne GNSS-IR, and satellite-based GNSS-R techniques, depending on the location of its GNSS receiver. It has been shown that the effective sensing area of ground-based GNSS-IR reaches at least 120 $m^2$ and can reach more than 1000 $m^2$ by combining multiple satellite tracks. It effectively monitors soil moisture from 0 to

5 cm in depth, achieving high accuracy inversion from bare soil to vegetation cover [28,29]. However, the relatively sparse distribution of GNSS stations eventually leads to the inability to achieve spatial continuity in soil moisture monitoring using ground-based GNSS-IR technology. In terms of airborne GNSS-IR, Sánchez et al. measured ground soil moisture by airborne GNSS-IR technique; it was jointly analyzed with maps with a high spatial resolution of reflectance, surface temperature, and digital surface models, and experiments showed that topography has an important influence on GNSS-IR signals [30]. Castellvi et al. combined hyperspectral imagery and airborne GNSS-IR technique inversion of soil moisture; a comparison with Airborne radiometer at L-band (ARIEL) soil moisture estimation was performed to obtain a high-resolution soil moisture product [31]. However, there are flight limitations and a relatively small range due to the airborne GNSS-IR technique. For this reason, the satellite-based GNSS-R technology, which loads GNSS receivers on a small satellite constellation for soil moisture monitoring, was developed [32]. It has become a hot topic of current research because of its advantages of high spatial resolution and low revisit time. Kim et al. developed a relative signal-to-noise ratio (rSNR) for deriving terrestrial soil moisture based on satellite-based GNSS-R; combining the rSNR with soil moisture values from SMAP gives daily soil moisture estimates [33]. Clarizia et al. also used the reflectance provided by satellite-based GNSS-R, combined with the auxiliary vegetation and roughness provided by the SMAP mission information to give daily soil moisture estimates in a grid with a resolution of 36 km × 36 km [34]. In addition to this, a few authors have replaced traditional algorithms with machine learning methods. It also combines a small amount of auxiliary data to improve the spatial and temporal resolution of soil moisture estimation. Fernández et al. proposed an algorithm to train a neural network using measured data to invert soil moisture from SMOS observations, and experiments showed that the neural network is an effective nonlinear regression tool [35]. Eroglu et al. proposed an artificial neural network-based method to retrieve daily soil moisture; soil moisture data from ground-based GNSS-IR and other auxiliary data, including normalized difference vegetation index (NDVI), vegetation water content (VWC), terrain elevation, terrain slope, and h-parameter (surface roughness) were input into the model modeling; finally, they obtained daily soil moisture estimates in a 9 km × 9 km grid [36]. Yuan et al. used neural networks to invert soil moisture using point-surface fusion and combined SMAP and multiple in situ observed soil moisture data based on generalized regression neural networks to build a soil moisture estimation model, ultimately improving the accuracy of the 9 km product of the SMAP task [37]. A follow-up study overcame the scale mismatch problem caused by a small spatial extent based on the triple configuration technique and used neural networks to combine bright temperature data from SMAP and other auxiliary data to build soil moisture estimation models [38]. Cui et al. combined soil moisture data from the Fengyun-3B satellite with surface temperature, normalized difference vegetation index, albedo, digital elevation model based on generalized regression neural networks, longitude, and latitude; finally, they improved the spatial and temporal resolution of the Fengyun-3B satellite from 0.25° and 2–3 days to 0.05° and one day [39]. However, most of these studies were based on the Spatio-temporal resolution of existing products, either improving the spatial or temporal resolution of the dataset, relying too much on the Spatio-temporal resolution of existing satellite-based GNSS-R products without comprehensive improvement. Also, the influence of surface environmental elements is not fully considered in using auxiliary data, such as rainfall, altitude, and some other vital factors that are not input.

In this paper, we propose a multi-data fusion learning method based on machine learning by combining ground-based GNSS-IR technology soil moisture data and surface environmental data. A multi-data fusion soil moisture model is constructed to obtain a spatially continuous soil moisture product of 500 m per day. We used surface environmental data, including (latitude and longitude information, NDVI, rainfall, air temperature, land cover type, and four topographic factors (elevation, slope, slope direction, and shading)). Since the above surface environmental data and soil moisture are related in a complex

non-linear manner, it is difficult to fuse multiple data types and map soil moisture using traditional linear statistical regression algorithms. Compared with traditional algorithms, machine learning techniques excel in dealing with complex non-linear problems. In particular, The Genetic Algorithm Back-Propagation Neural Network model optimized by the genetic algorithm (GA) is highly stable and well fitted. Therefore, we input the processed data into the trained GA-BP neural network model and finally obtained the soil moisture map of 500 m per day for 15 days during 15 February 2014–1 March 2014 for the western coast of the United States.

The rest of the paper is described below. Section 2 describes the ground-based GNSS-IR data, the post-validation data National Aeronautics and Space Administration and the U.S. Department of Agriculture (NASA-USDA) products, and the network structure of the GA-BP model. Section 3 describes the study area and the pre-processing process of soil moisture data and related geoenvironmental elements from GNSS-IR stations. Section 4 compares and analyzes the accuracy of the soil moisture products generated based on the GA-BP model and the method's feasibility. Section 5 gives the conclusion of the paper and provides an outlook for future research.

## 2. Materials and Methods

### 2.1. PBO Project

The National Science Foundation's (NSF) Plate Boundary Observatory (PBO), which began construction in 2004, was completed in 2008 [40]. The PBO is a core network of 1100 continuously operating GNSS stations. The network also contains 1 Hz and 5 Hz high sampling rate stations capable of observing millimeter changes in GNSS station locations over days to years. The essence of this is GNSS satellites transmitting signals, which are L-band microwave signals (~1.2 and ~1.5 GHz). The ground receiver antenna receives both direct and reflected signals, while the reflected signals vary with soil moisture, snow depth, and vegetation conditions. The changes in surface reflections are recorded in the signal-to-noise ratio (SNR) data, which is then solved to quantify soil moisture, snow depth, and vegetation growth rate. The network is the only one operating on the principle of GNSS-IR technology. Soil moisture data from the PBO project can be downloaded from the International Soil Moisture Network (ISMN) website and available on the PBO data portal. The basic parameters of all stations of the PBO network in the study area are shown in Table 1.

**Table 1.** Basic parameters of PBO measurement stations.

| Name | Parameters |
| --- | --- |
| Receiver Type | TRIMBLE NETRS GPS |
| Antenna Type | TRM29659.00 |
| Rectifier type | SCIT |
| Station height and sampling rate | 2 m, 15 s |
| Effective depth for measuring soil moisture | 0–5 cm |

### 2.2. NASA-USDA Soil Moisture Data

National Aeronautics and Space Administration Goddard Space Flight Center (NASA GSFC) provided the NASA-USDA global soil moisture data by the 1-D Ensemble Kalman Filter (EnKF) data assimilation method. SMOS level 2 soil moisture observations were generated by integrating them into a modified two-layer Palmer model [41]. Due to the low resolution of SMOS itself, as a result, the NASA-USDA global soil moisture data generated based on SMOS data has a low spatial resolution. The spatial resolution of this dataset is only $0.25° \times 0.25°$. This dataset includes both surface and subsurface soil moisture data, but the soil moisture based on GNSS-IR can only reflect the variation of soil moisture within 1–6 cm of the soil surface. Therefore, the surface soil moisture of the NASA-USDA global soil moisture data is selected as the initial comparison data for the point-surface fusion results. This paper obtained NASA-USDA global soil moisture through Google

Earth Engine (https://developers.google.com/earth-engine/datasets/catalog/NASA_USDA_HSL_soil_moisture#description, accessed on 6 March 2021) data.

### 2.3. GNSS-IR Technology for Inversion of Soil Moisture

SNR observations are the core computational data of GNSS-IR technology, an index describing the signal quality of GNSS antennas. It is mainly influenced by the combination of elements, such as receiver antenna gain, satellite altitude angle, and multipath effect [42].

The GNSS receiver receives both direct and reflected signals from GNSS satellites. The continuous movement of GNSS satellites makes the GNSS direct reflection signal constantly change, which makes the characteristic parameters of the interference waveform constantly change over time, and ordinary geodetic receivers will record these changes in the form of SNR [43]. Therefore, the study of SNR can estimate soil moisture through the change of the characteristic parameters of the interference effect. The ground multipath error model is shown in Figure 1.

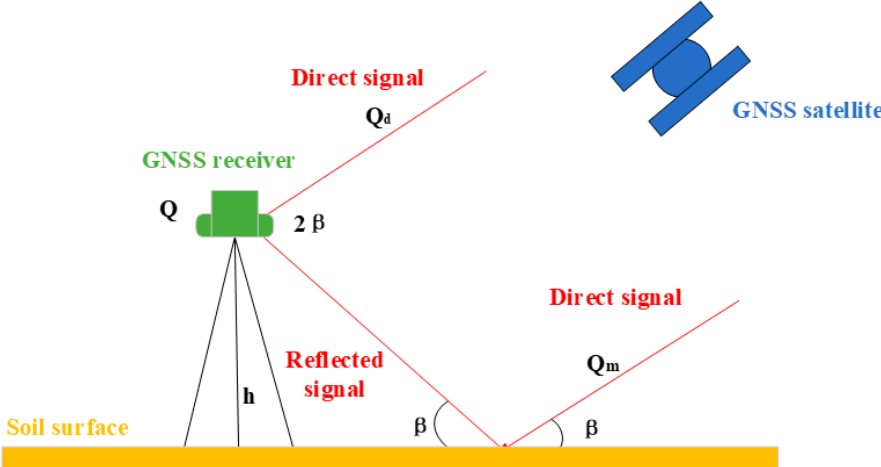

**Figure 1.** Ground multipath error model.

From Figure 1, the phase difference between the direct and reflected signals of the GNSS satellite can be deduced as:

$$\alpha = \frac{4\pi h}{\lambda} \sin\beta \tag{1}$$

where $h$ denotes the vertical height of the GNSS receiving antenna from the ground and $\beta$ denotes the angle $\lambda$ between the GNSS signal and the ground surface is the L carrier wavelength. Further study reveals that the SNR can be expressed in terms of direct and reflected signals as follows.

$$SNR^2 = Q_d^2 + Q_m^2 + 2Q_d Q_m \cos\alpha \tag{2}$$

In Equation (2), $\alpha$ denotes the phase difference between the direct and reflected signals of GNSS satellites, and $Q_d$ and $Q_m$ denote the amplitudes of the direct and reflected signals of GNSS satellites, respectively. Chew et al. [44] found that after removing the direct signal amplitude $Q_d$, only the reflected signal amplitude $Q_m$ is retained in the SNR observations. There is a certain sine or cosine relationship between $Q_m$ and sinβ, which can be expressed as:

$$SNR_D = Q_m \left( \frac{4\pi h}{\lambda} \sin\beta + \varphi_m \right) \tag{3}$$

In Equation (3), $\varphi_m$ denotes the relative delayed phase. In the case where $SNR_D$ is known, we can use the Lomb-Scargle spectral analysis transform to find the frequency $\frac{4\pi h}{\lambda}$ and then solve for the magnitude $Q_m$ and the relative phase $\varphi_m$ using a least squares fit.

A strong correlation between $\varphi_m$ and soil moisture values was found by Chew et al. [45], which is the inverse the best parameter for inversion of surface soil moisture.

Based on this, Chew et al. [46] smoothed phase $\varphi_m$ using a moving average filter to remove the expected phase variation due to vegetation and add the residual water content in the soil ($SMC_r$) to produce a phase $\varphi_{sm}$ that reflects only the variation in soil moisture. The phase $\varphi_{sm}$ was then related to soil moisture $SM$ is expressed as follows:

$$SM = S_{\varphi_{sm}} + SMC_r \qquad (4)$$

In Equation (4), $S$ is the expected slope (between soil moisture and phase), $\varphi_{sm}$ indicates the phase change due to soil moisture, and $SMC_r$ is available through public data [47].

*2.4. Data Pre-Processing*

The objective of the method is to obtain soil moisture products with a high spatial and temporal resolution by fusing ground-based GNSS-IR data with surface environmental parameters extracted from optical remote sensing. The specific process of the multi-data fusion model in this study is described below, and the flow chart of the method is shown in Figure 2.

(1) Data processing. Download the soil moisture retrieved by GNSS-IR technology through the International Soil Moisture Network (ISMN). Use Google Earth Engine (GEE) to obtain image data of surface environmental elements (latitude, longitude, NDVI, temperature, rainfall, land cover type, slope, aspect, elevation, and shadow) of the experimental area (1 January 2014–1 March 2014). GEE's image pyramid strategy specifies the output image with a spatial resolution of 500 m and a temporal resolution of 1 day.

(2) Build the data set. According to each GNSS station's latitude and longitude information, the corresponding image value of each GNSS station is extracted. Ten surface environment elements are used as the input of the GA-BP neural network model to form the input data set. Take GNSS-IR soil moisture as the training target (output data) to construct an output data set. This makes the input layer of the GA-BP neural network have 10 neurons, while the output layer has only one neuron.

(3) Model building. Import the modeling input data set and the modeling input data set into Matlab, and divide all the data into 70%, 15%, and 15% as the training set, validation set, and test set for model construction. Use the divided training set and confirmation set to train the GA-BP neural network model, and use the test set to test the accuracy of the trained model. Save the GA-BP neural network model (trained qualified neural network) whose accuracy reaches the threshold.

(4) Accuracy verification. First, the reliability of the neural network model that reaches the threshold is tested by the tenfold cross-validation method. Secondly, a verification input data set formed 10 kinds of surface environment elements corresponding to the GNSS stations not involved in the modeling. Input the validation data set into the trained neural network model and output the GA-BP inversion soil moisture data set. The GA-BP inversion soil moisture data set is compared and analyzed with the GNSS-IR soil moisture corresponding to the stations not involved in the modeling. If the accuracy meets the requirements, the neural network model trained to reach the threshold is reliable and effective.

(5) Production soil moisture map. Each 500-m square in the experimental area corresponds to a latitude and longitude coordinate, and 10 kinds of surface environmental elements corresponding to all latitude and longitude coordinates in the experimental area are extracted through the latitude and longitude coordinates to form a map input data set. Input the mapped input data set into the GA-BP neural network model that reaches the threshold and obtain the GA-BP inversion soil moisture data set for mapping through the training output. Import the GA-BP inverted soil moisture data set used for mapping into ArcGIS, and use ArcGIS "Point to raster" function to convert all the GA-BP inverted soil moisture data sets used for mapping into raster images (Soil moisture map).

(6) Soil moisture map verification. The soil moisture map (500 × 500 m) and NASA-USDA (0.25° × 0.25°) products, NDVI (500 × 500 m), and rainfall (500 × 500 m) were compared and analyzed. Due to the different units and resolutions of the four Same. We only analyze whether the generated soil moisture map is qualified by changing the map spots and the value between the same areas. On this basis, we extracted 392 sites based on the latitude and longitude of each NASA grid center and analyzed the correlation between the NASA soil moisture at the sites and the soil moisture retrieved by GA-BP. Further, evaluate the performance of GA-BP inversion of soil moisture.

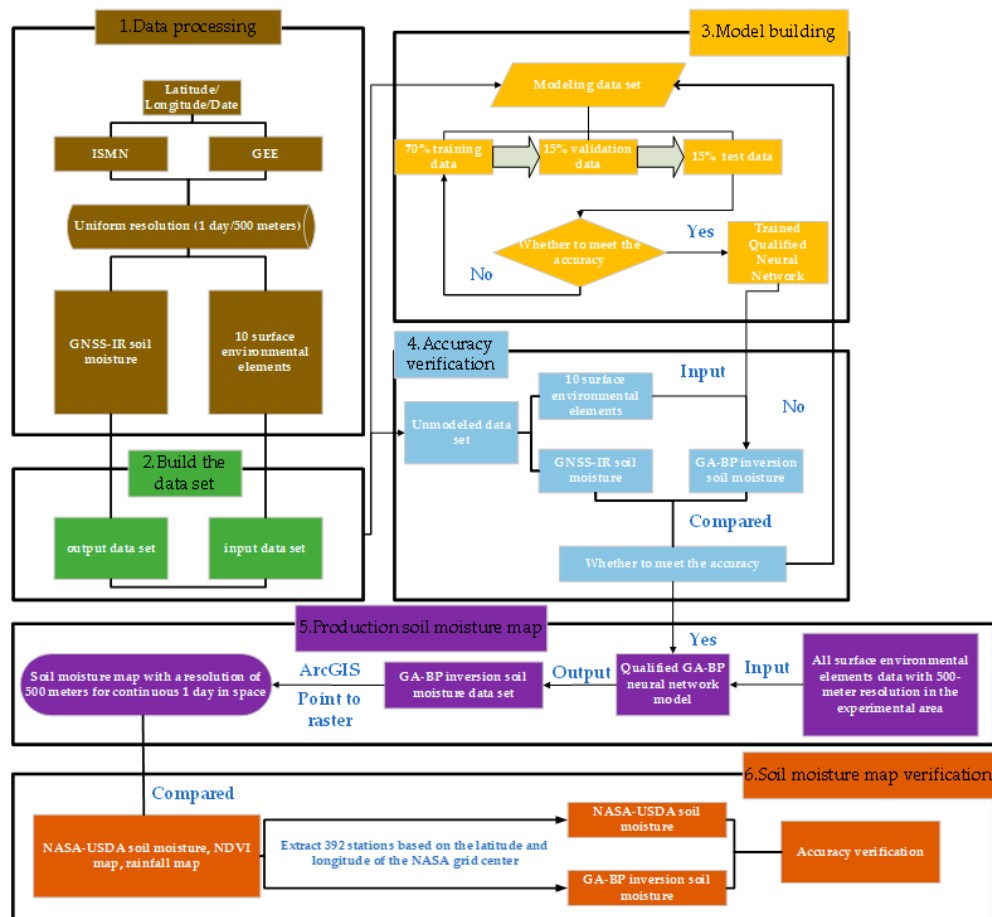

**Figure 2.** Flow chart of multi-data fusion model.

## 2.5. GA-BP Neural Network

### 2.5.1. BP Neural Network

Backpropagation (BP) neural network is a multilayer feedforward network model consisting of two processes: forward propagation of information and backward propagation of error [48]. It is a more widely used neural network with solid adaptability and learning ability and can better solve nonlinear problems. Its essence is learning by stochastic gradient descent solving algorithm. The input and output layers output data are called forward propagation, called backward propagation, using the weights and deviations calculated in each layer to update the model for iteration. BP neural network mainly consists of the input layer, hidden layer, and output layer. Its network structure is shown in Figure 3. Our study's input signals are latitude, longitude, NDVI, rainfall, air temperature, land cover type, and four topographic factors (elevation, slope, slope direction, and shading); the output parameter is GNSS-IR soil moisture. BP neural network model is implemented by using the neural network toolbox of MATLAB. The number of neurons m in the implicit layer of the BP neural network takes values between $\sqrt{2n} + 1$ and $2n + 1$. n is the number of neurons in the input layer, so we tested hidden layer neurons ranging from 7 to 21.

Using a 10-fold coefficient performs best by validating that the best model performance is obtained when the number of hidden layer neurons is 19. Also, set the number of training steps for the BP neural network to 1000, the training accuracy to 0.001, and the learning rate to 0.0001.

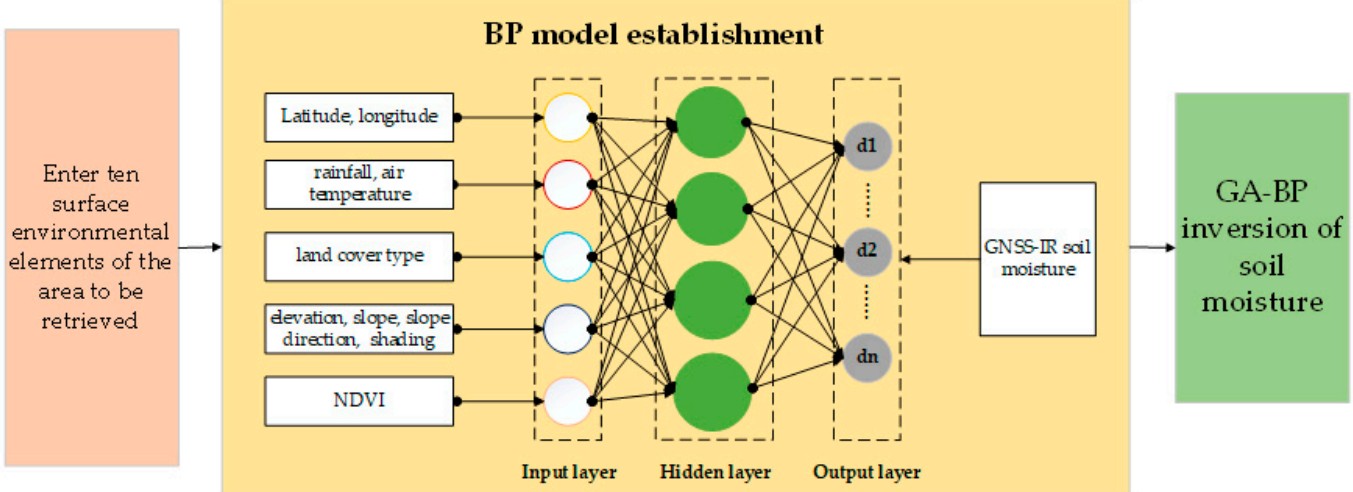

**Figure 3.** Structure of GA-BP neural network.

However, the number of neurons in the hidden layer of the BP neural network needs to be manually selected, and the weights and thresholds are also randomly generated, which causes the BP neural network to converge slowly and quickly fall into a local minimum [49]. For this reason, this paper uses the GA algorithm to optimize the BP neural network.

### 2.5.2. The Genetic Algorithm

The GA algorithm is a global search computer algorithm derived mathematically based on inheritance laws in nature [50]. Its essence is selecting good individuals from the population; through crossover and mutation operations to obtain new individuals of good quality. The advantages of the genetic algorithm: (1) It can quickly search the whole solution in the solution space and has excellent global search capability. (2) It is suitable for distributed computing, and natural parallelism speeds up the convergence speed. (3) Simple, general, and wide range of applications. The GA algorithm is used to implement the optimization of BP neural network weights. The adaptive adjustment of the crossover and variance probability enables individuals to update the network weights continuously, thus improving the BP neural network's network convergence speed and algorithmic accuracy. For this reason, this paper uses the GA algorithm to optimize the BP neural network. The global search property of the GA algorithm is combined with the powerful nonlinear learning ability of the BP neural network to improve the training ability of the model. The BP neural network optimized by the GA algorithm is called the Genetic Algorithm Back Propagation neural network. The calculation process of the whole GA-BP neural network is shown in Figure 4.

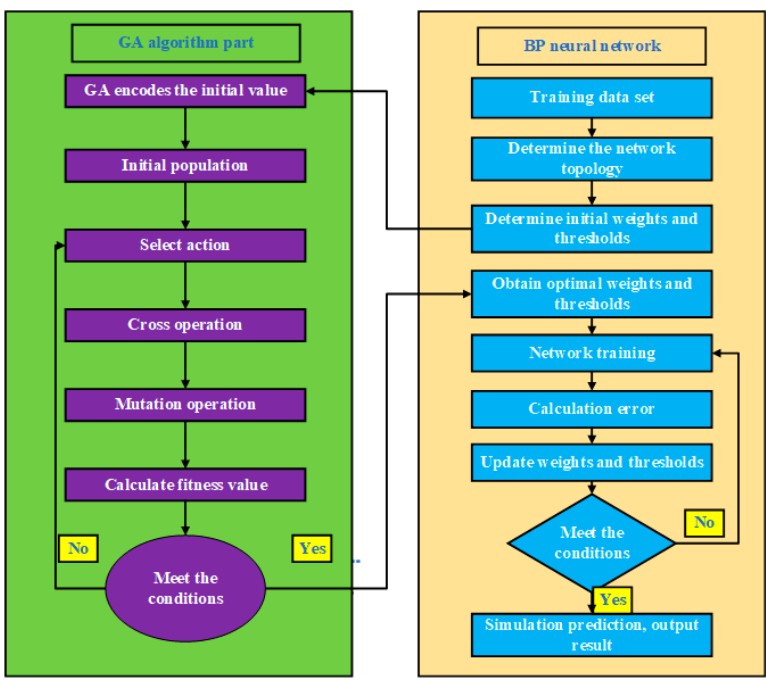

**Figure 4.** GA-BP model calculation process.

The content of the GA-BP neural network is divided into two parts: On the one hand, the GA algorithm is used to globally search for the optimal solution to find a set of optimal solutions. On the other hand, the optimal global solutions are used as the initial weights of the BP neural network. In this study, the initial population size of the GA algorithm is set to 50, the number of genetic generations to 100, the crossover probability to 0.3, and the variance probability to 0.09.

### 2.6. Validation Method and Evaluation Metrics

This study applied a 10-fold cross-validation technique to test the model overfitting and predictive ability [51]. The training and validation data were run in 10 random iterations. In each iteration, the entire data set was randomly divided into ten equal-sized portions. One of the copies is used as the validation sample, and the remaining nine copies are used as the training sample for one iteration. In the next iteration, one of the previous training samples is used as the validation sample, and the remaining nine are used as the training samples in the next iteration. Repeat this step nine times until ten iterations are completed, and we will get the prediction results for the whole data set of soil moisture. The model cross-validation results are obtained by averaging the ten results. These averaged cross-validation results provide a good check of whether the model is overfitted. When the model is poor, the cross-validation results are also poor, and the model with the most significant correlation coefficient is selected as the best-fit model for subsequent predictions.

To verify the validity of each model, we quantitatively evaluated the training and test sets using the Pearson correlation coefficient R, root mean square error (RMSE), unbiased root mean square error (ubRMSE), and mean bias (bias). R describes the degree of model convergence between +1 and 1. Where +1 indicates a perfect positive linear correlation, 0 indicates no linear correlation, and 1 indicates a perfect negative linear correlation. RMSE and bias measure the deviation between the inverse soil moisture values and the measured values. RMSE of 0 indicates no deviation. The bias of 0 indicates an unbiased estimate, more remarkable bias than 0 is an overestimate, and less than 0 is an underestimate. In general, the smaller the two, the better. ubRMSE is the random error. ubRMSE eliminates possible additional bias when the measured value is considered the actual value, and the smaller the value is, the better the model performance is.

## 3. Study Area and Data

### 3.1. Study Area

Considering that the dense distribution of the stations affects the modeling effect in local areas [52]. Most GNSS stations are deployed in the western coastal region, and the distribution is denser. In this study, we select the western part of the continental United States (32° N–39° N, 114° W–123° W) as the study area. We obtained soil moisture values for a total of 60 days from 1 January 2014 to 1 March 2014 through The International Soil Moisture Network (ISMN) [53] (https://ismn.geo.tuwien.ac.at/en/, accessed on 6 March 2021) for the experimental study. Here are 50 stations in the experimental area, considering the density of GNSS station distribution and land cover type. In this paper, 44 stations (brown circles) were selected to participate in the point-surface fusion modeling training; 6 stations (gray circles) were selected as test stations, which were not involved in the modeling and only used as the last results verification. Figure 5a,b show the GNSS stations and the distribution of land cover types in the selected experimental area, respectively.

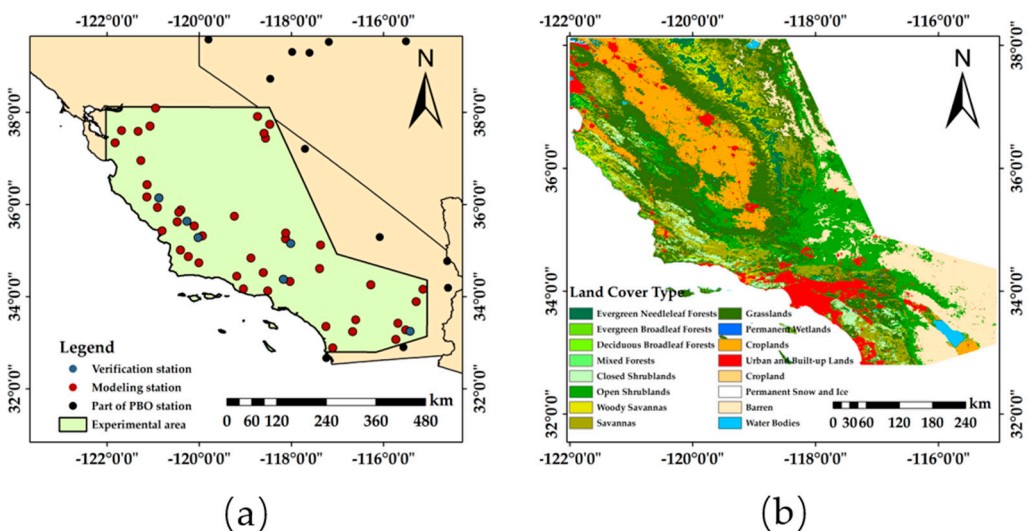

**Figure 5.** Overview of the study area for this study (32° N–39° N, 114° W–123° W), (**a**) experimental area in green, modeling stations in brown, and validation stations in gray; (**b**) land cover type of the experimental area.

As seen in Figure 5b, the primary topography of the region is high mountains and plateaus. The central and northern parts of the test area are mostly evergreen coniferous forests as well as grasslands. The central part is the central valley, which is cultivated mainly by agricultural land. The left side of the central valley is the Coastal Range; the right side is the Sierra Nevada, which is dominated by evergreen coniferous forests and savannas. The southeastern part of the test area is barren, primarily desert and open sagebrush. It can be seen that each direction of the experimental area has unique geomorphological features and the geographical conditions are pretty different. In terms of climate, the western coastal region of the test area has a Mediterranean climate, the Sierra Nevada region has a highland mountain climate, and the southern and southeastern regions have a tropical desert climate. Although there are multiple climates in the test area, the overall Mediterranean climate is still present due to the overall proximity to the ocean. In terms of rainfall, the northwestern part is near the coast and receives more rainfall. The central valley is a mountainous area with a high elevation and less rainfall. The southeastern desert region has no rainfall. Thus, it can be seen that the soil moisture in the test area will have significant differences both in space and time, and the experiment has strong feasibility.

### 3.2. Other Geographic Auxiliary Data

To ensure the comprehensiveness and reliability of the soil moisture inversion model, we used the surface environmental factors around the measurement stations as input data sets. These data are (1) Latitude and longitude information, calculated by ArcGIS based on World Geodetic System-1984 (WGS-84) Coordinate System. (2) Land cover type, extracted from the International Geosphere-Biosphere Program (IGBP) land cover map based on Moderate Resolution Imaging Spectroradiometer (MODIS) to obtain land cover class data [54]. (3) normalized difference vegetation index, based on the MODIS water surface reflectance daily global 500 m dataset [55], was calculated from the near-infrared (NIR) and infrared (RED) bands of the reflectance data. The formula is NDVI = (NIR − RED)/(NIR + RED). We used a sliding window of more than 16 days on average, centered on calculating the NDVI for a particular day (eight days ahead and seven days past), obtaining the NDVI value with slight fluctuation in short periods. (4) Topographic factors, which are usually used to assess the influence of coarse surface over topography on soil moisture, are accessed through the Google Earth Engine for the dataset [56]. Topographic data (elevation, slope, slope direction, and shading) from the NASA Shuttle Rader Topography Mission Digital Elevation 30m dataset were extracted with the help of the extremely high computing power of the Google Cloud Platform. (5) Rainfall and air temperature are critical meteorological parameters for the surface environment and climate change and also play an essential role in vegetation growth and have a more substantial influence on soil moisture [57]. We obtained the US region's daily average air temperature and rainfall through the Phase Rotated Intense Slow Moonbeam (PRISM) project [58]. (7) NASA-USDA global soil moisture data generated based on SMOS data have a low spatial resolution and include surface and subsurface soil moisture data. In this paper, we select the surface soil moisture of NASA-USDA global soil moisture data as the preliminary comparison and validation data for the point-surface fusion results. All the data mentioned above are available for download through the Google Earth Engine, and the specific products used in this study are listed in Table 2 below.

**Table 2.** Auxiliary data used in the study.

| | Environmental Parameters | Spatio-Temporal Resolution | Project | Time |
|---|---|---|---|---|
| Auxiliary geographic environment data | NDVI | 1 day/500 m | MOD09GA | 1 January 2014–1 March 2014 |
| | Land cover type | 1 day/500 m | MCD12Q1 | 2014 |
| | Rainfall, air temperature | 1 day/2.5′ | PRISM | 1 January 2014–1 March 2014 |
| | elevation, slope, slope direction, shading | 30 m | NASA SRTM Digital Elevation 30 m | 2000 |
| Validation data | NASA-USDA soil moisture | 3 days/0.25° | NASA GSFC | 1 January 2014–1 March 2014 |

The above table shows that the maximum temporal resolution for all surface environmental parameters is one day. Meanwhile, the Google Earth Engine sets the spatial image resolution to 500 m when outputting images with specified spatial resolution according to the image pyramid strategy. Therefore, this study's final multi-data fusion of soil moisture products has a spatial and temporal resolution of 500 m per day.

## 4. Experiment and Analysis

### 4.1. Modeling

We input the processed input dataset (latitude and longitude information, rainfall, air temperature, land cover type, NDVI and four topographic factors (elevation, slope, slope direction, and shading); and the output dataset (ground-based GNSS-IR soil moisture data) into the GA-BP neural network for iterative training. We obtained the optimal GA-BP multi-data fusion soil moisture model after 300 training cycles. Figure 6. shows scatter plots of the Inversion values and GNSS-IR soil moisture actual values for the entire data set of the optimal model and the test set, along with Pearson correlation coefficient R values, RMSE, bias, and ubRMSE. In addition, Table 3 shows the accuracy statistics of the modeled measurement stations.

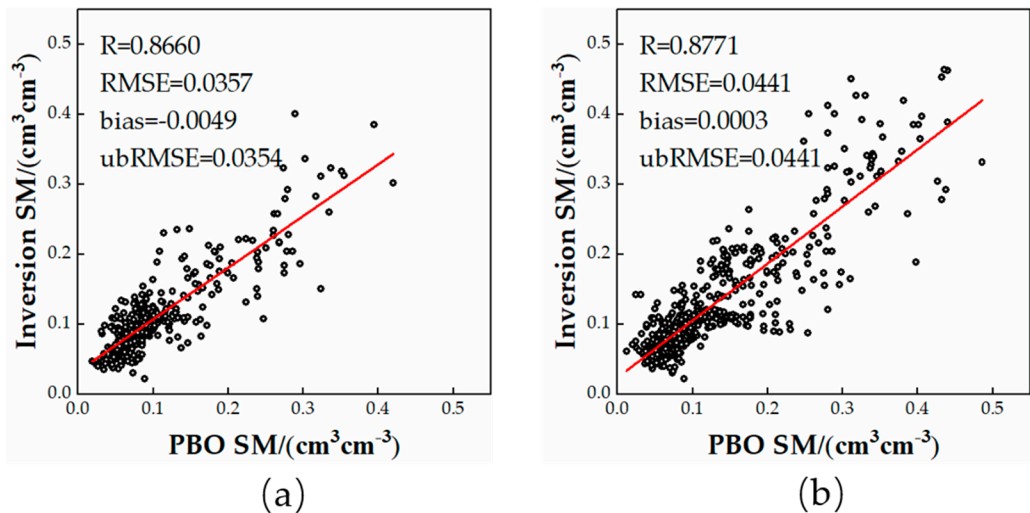

**Figure 6.** (**a**) Scatterplot of random sampling retrieval for the whole dataset; (**b**) Scatterplot of the test set data (15 February 2014–1 March 2014) for 44 modeling stations, providing Pearson correlation coefficients R, RMSE, bias, and ubRMSE.

**Table 3.** Accuracy statistics of modeled measurement stations including R, RMSE, bias, and ubRMSE.

| Accuracy Index | Scope | Number of Measuring Stations | Average |
|---|---|---|---|
| R | 0–0.4 | 5 | |
| | 0.4–0.6 | 3 | 0.8967 |
| | 0.6–0.1 | 36 | |
| RMSE | <0.04 | 29 | |
| | 0.04–0.06 | 10 | 0.0408 |
| | >0.06 | 5 | |
| bias | <0.02 | 22 | |
| | 0.02–0.03 | 16 | 0.0002 |
| | >0.03 | 6 | |
| ubRMSE | <0.04 | 33 | |
| | 0.04–0.06 | 9 | 0.0407 |
| | >0.06 | 2 | |

It was considering that the Pearson correlation coefficient R takes the values of [0–0.4], [0.4–0.6], and [0.6–1] as weak, medium, and strong correlations, respectively. The above table shows that 33 stations with Pearson correlation coefficients greater than 0.6 are strongly correlated, accounting for 75% of all stations. 3 stations are moderately correlated, and the remaining 18.1% are weakly correlated. The overall correlation of all stations reached 0.8770, and despite the poor performance of some stations, a nonlinear regression of the entire data was very effective. Like the correlation, the SMAP task was produced

with a precision of 0.04 due to its accuracy. Therefore, RMSE below 0.04 is considered high performance, [0.04–0.06] as medium performance, and greater than 0.06 as more unreasonable performance. In this case, there were 28 stations with RMSE less than 0.04, accounting for 63.6% of all stations. We are further combined, with the ubRMSE, it can be seen that there are 30 stations with ubRMSE less than 0.04 (high performance), accounting for 68.1% of all stations. Based on the four of the above table, it can be seen that the short delay estimation of our algorithm seems to be consistent with the average short delay level of most stations. The vast majority of stations achieved good modeling results, except for a few stations that did not meet expectations.

### 4.2. Model Validation

In order to test the feasibility of the above-trained GA-BP multi-data fusion soil moisture inversion model to retrieve soil moisture in areas other than modeling, we input 10 surface environmental elements from six specific sites that were not involved in modeling. In the trained GA-BP neural network model, the output is GA-BP inverted soil moisture. The GA-BP retrieved soil moisture was compared with the GNSS-IR soil moisture of the characteristic site. Four indicators (Pearson correlation coefficient R, RMSE, bias, and ubRMSE), were used to analyze their accuracy. The results are shown in Table 4.

**Table 4.** Accuracy statistics of the six unmodeled measurement stations.

| Test Stations | R | RMSE | Bias | ubRMSE | Land Cover Type |
|---|---|---|---|---|---|
| ANGUS_PROP | 0.4040 | 0.0403 | −0.0392 | 0.0092 | Barren |
| CALCITYAPT | 0.8824 | 0.0331 | 0.0273 | 0.0188 | Open bushes |
| CARRIZORAN | 0.9516 | 0.0490 | 0.0141 | 0.0470 | Grassland |
| MOONEYCYN | 0.8199 | 0.0734 | −0.0635 | 0.0367 | Grassland |
| MT_GLEASON | 0.8800 | 0.0643 | 0.0170 | 0.0620 | Savanna |
| WICKSRANCH | 0.8342 | 0.0331 | 0.0071 | 0.0324 | Grassland |

As shown in Table 4, five Pearson correlation coefficients R are more significant than 0.6 for the six test stations. The highest correlation is 0.9329, which has a robust correlation. The worst correlation is only 0.0143 for the stations in barren areas. Only one station has an ubRMSE greater than 0.06, and four stations are less than 0.04. It can also be seen that the stations with the land cover type of grassland have better inversions and fewer errors in flat areas, and there are no gross errors. In addition to the rainfall factor, the model is influenced by NDVI due to its significant influence. For this reason, in areas with no vegetation cover, desert, there is no vegetation. It also means that there is no reasonable NDVI, which influences the final inversion results. Although the test results of one or two stations did not meet the expectations, and the errors were relatively large. However, the vast majority of the tested stations achieved good results with minor errors. This shows that the trained GA-BP multi-data fusion soil moisture inversion model is feasible and accurate for inverting soil moisture in areas other than modeling.

In order to verify the correlation between NASA-USDA soil moisture, GNSS-IR soil moisture, and GA-BP model inversion of soil moisture. We compare and analyze the NASA-USDA soil moisture corresponding to the latitude and longitude of the 50 modeling stations, the GNSS-IR soil moisture, and the soil moisture retrieved by the GA-BP model. Because the units of the three are different, the error of the three cannot be calculated. Therefore, this paper only uses the Pearson correlation coefficient R to verify, and the comparison results are shown in Table 5.

**Table 5.** Comparison of soil moisture values for NASA-USDA, PBO, and GA-BP models.

| Accuracy Index | Pearson's Correlation Coefficient R | | | |
|---|---|---|---|---|
| NASA-USDA and PBO | <0.6<br>12 | 0.6–0.8<br>7 | >0.8<br>31 | Average<br>0.8222 |
| NASA-USDA and GA-BP inversion values | <0.6<br>14 | 0.6–0.8<br>6 | >0.8<br>30 | Average<br>0.8471 |

As seen in Table 5, 39 stations out of 50 with Pearson correlation coefficient r more significant than 0.6 between NASA-USDA soil moisture values and GNSS-IR based inversion of soil moisture, accounting for 78% of the total number stations. Meanwhile, among these stations, less than 0.6, the NASA products of 7 stations are unchanged during the inversion period, resulting in poor inversion results. The actual number of stations with fluctuations is 43, which shows that the actual effect of our algorithm is better than expected. There are 31 stations that correlate greater than 0.8, accounting for 72% of the fluctuating stations. The correlation between the soil moisture and NASA-USDA soil moisture values obtained from the inversion of this paper is strong, with 40 stations correlating 0.6, accounting for 80% of the total number of stations. There are 34 stations that correlate greater than 0.8, with an average correlation of 0.7770. This shows that the GA-BP model in this paper is accurate and consistent with the NASA-USDA soil moisture results, and the accuracy is better than GNSS-IR soil moisture. Moreover, the resolution of the inversion of soil moisture by the GA-BP model in this paper is higher than that of NASA-USDA soil moisture data, which can better represent the differences and changes in soil moisture in the region. Therefore, it can be shown that the fusion of multiple data of soil moisture using the neural network proposed in this paper is feasible, and the fusion model based on GA-BP neural network established in the previous paper is accurate and effective.

*4.3. 500 m Daily Soil Moisture Map Generation*

Due to the good predictive ability of the GA-BP model to retrieve soil moisture, we combined all 10 surface environmental elements in the $500 \times 500$ squares in the experimental area into a map input set and input the qualified GA-BP neural network model trained above to obtain Continuous soil moisture products in 500 m of space per day. Figure 7. Comparison of the 15-day GA-BP inversion soil moisture map and the 5-day NASA-USDA soil moisture product from 15 February 2014 to1 March 2014.

From Figure 7((a1)–(a4),(b1)–(b4),(c1)–(c4),(d1)–(d4),(e1)–(e4)) contrast can be seen. The soil moisture map retrieved by GA-BP is consistent with the spatial distribution of NASA-USDA soil moisture products. In the area with high NASA-USDA soil moisture value, the soil moisture value retrieved by GA-BP is also high, the soil moisture value retrieved by NASA-USDA is low, and the soil moisture value retrieved by GA-BP is also low. However, the soil moisture map of NASA-USDA represented by (a4), (b4), (c4), (d4), and (e4) has a low spatial resolution, which can not distinguish the terrain of the experimental area, let alone show the change of soil moisture in each area in detail. It can only reflect the overall, low-resolution soil moisture variation in the experimental area, and it is challenging to distinguish soil moisture variation in different areas. It can only show a whole patch of the same color, and the temporal resolution is low, which cannot reflect the daily changes of soil moisture. In contrast, the soil moisture product generated in this paper provides daily soil moisture changes over a 500-m range. Under the same climatic conditions, whether it is the Central Valley floor in the center, the Coastal Mountains in the west, the Sierra Nevada in the south, or the desert areas in the southeast, it can show the differences in soil moisture in these regions in more detail.

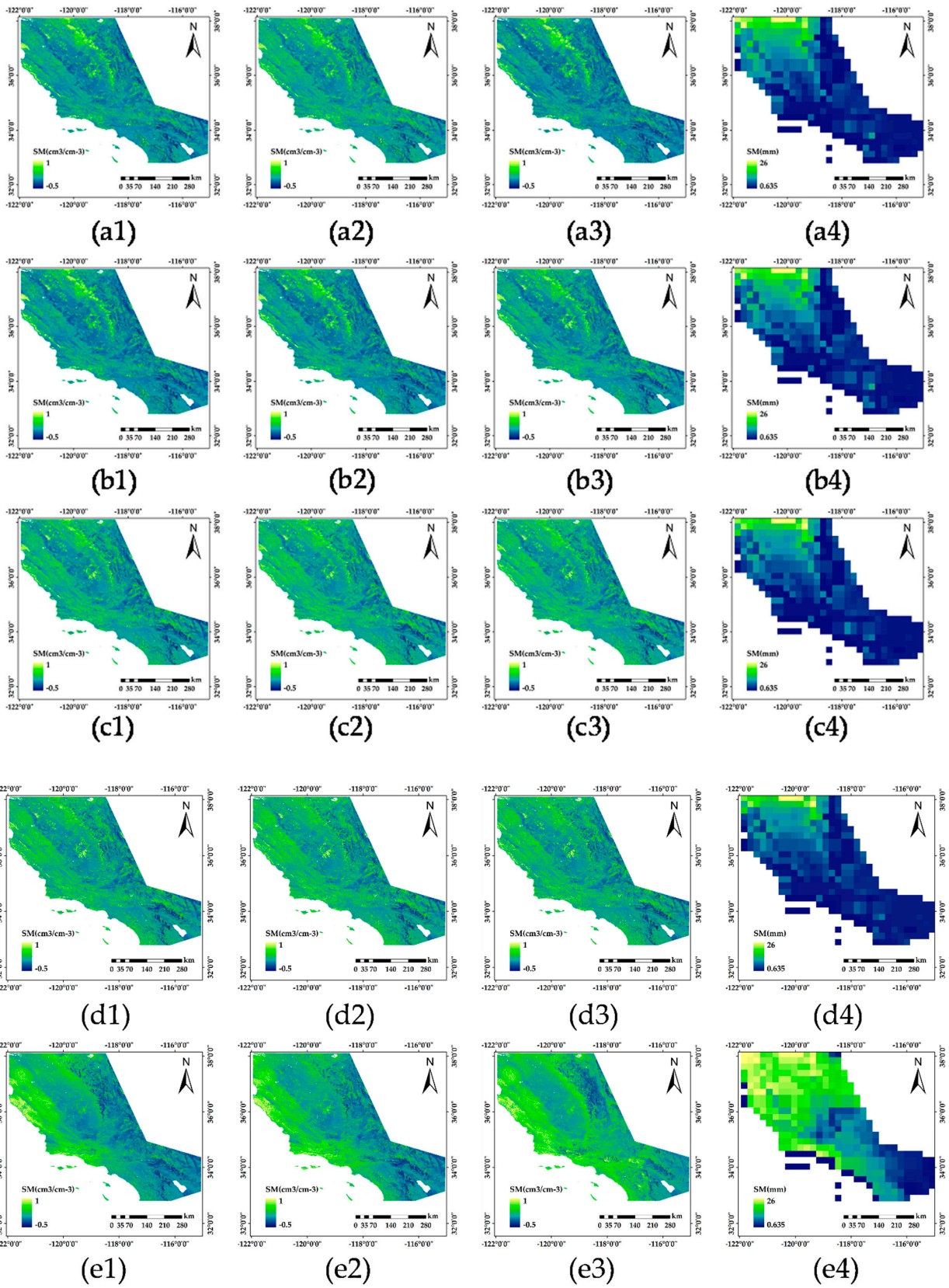

**Figure 7.** (**a1**–**a3**,**b1**–**b3**,**c1**–**c3**,**d1**–**d3**,**e1**–**e3**) are 15 February 2014–1 March 2014 GA-BP inverted soil moisture map for 15 consecutive days during the period, (**a4**,**b4**,**c4**,**d4**,**e4**) correspond to the NASA-USDA soil moisture during the period 15 February 2014–1 March 2014 In the picture, the time resolution of NASA products is 3 days, so there are only 5 soil moisture maps in 15 days, one every three days. Blank areas were removed as ancillary data for water body impact areas.

The (a4), (b4), (c4), (d4) of the NASA-USDA products show no considerable variation in soil moisture during 15 February 2014–26 February 2014. This is basically consistent with the GA-BP inverted soil moisture map ((a1)–(a3), (b1)–(b3), (c1)–(c3), (d1)–(d3)) generated in this study. During this period, the soil moisture in each region did not change much. The northeastern region is primarily grassland and evergreen coniferous forest with denser vegetation cover. The southeastern part is a coastal area near the sea. The central part is a predominantly agricultural area with year-round irrigation and more rain, resulting in high soil moisture values in the areas mentioned above. The southwest is mostly a desert area with little rainfall all year round, so the soil moisture values are low. In terms of time, (e1)–(e3) is compared with (a1)–(a3), (b1)–(b3), (c1)–(c3), (d1)–(d3), and it can be seen that in most areas the soil moisture have risen drastically. This is due to three consecutive days of continuous rainfall in the eastern coastal and central regions during 27 February 2014–1 March 2014, which significantly increased the soil moisture value. At the same time, during this period, the soil moisture map (e4) of NASA-USDA products also increased significantly compared to (a4), (b4), (c4), and (d4). The soil moisture increase area in the soil moisture map (e4) of NASA-USDA products is the same as the soil moisture map (e1)–(e3) retrieved by GA-BP. This preliminarily shows that the soil moisture map retrieved by GA-BP in this paper is accurate.

In order to further verify the reliability of the soil moisture map retrieved by GA-BP in this study, we compared rainfall, NDVI map, and soil moisture. Existing studies have shown that NDVI can reflect the vegetation coverage information and reflect the soil moisture information under the vegetation coverage state [59]. In addition, precipitation can be said to be the first influencing factor in bare-soil environmental areas. It also plays a vital role in vegetation growth and significantly impacts soil moisture [60]. Figure 8. shows the rainfall, NDVI, and GA-BP inverted soil moisture map from 27 February 2014 to 1 March 2014.

As seen in Figure 8(a1)–(a3), the NDVI in this region, which fluctuates up and down in a small range, does not vary significantly. However, because NDVI is calculated by waveband, the optical image information is highly susceptible to interference. Meanwhile, in the desert and urban building areas without vegetation cover, the NDVI calculation is seriously affected because of the low vegetation cover. This results in NDVI values greater than 1 or less than 1 in a small number of places, making negative soil moisture values in these areas. In later experiments, this study will be improved for different NDVI calculations in these different areas. The distribution of NDVI in the region is also generally consistent with the land cover type map in Figure 5b. The southwestern part is a desert, barren area with basically no vegetation cover. For this reason, the NDVI values are low, which also makes the soil moisture values in this region poorer compared to other regions. The northeastern and central areas are primarily arable land, surrounded by arable land as grassland; more peripheral are tall shrub-like vegetation and coniferous forests, resulting in higher NDVI values in the eastern coastal and central areas of the test area. (b1)–(b3) shows the rainfall distribution during the period, as we can see. The central-eastern coastal area had substantial rainfall on 27 February 2014 (red box range), as rainfall was directly and strongly influenced. The soil moisture model of this experiment captures this vital information better, and the soil moisture values in the red-boxed area of the GA-BP Inversion soil moisture map (c1) on that day have increased significantly. The continuous rainfall at the exact location from 28 February 2014–1 March 2014 has further increased the soil moisture values in this area. Our GA-BP Inversion soil moisture map all reflects the soil moisture changes in these areas better. From the red boxed areas of the GA-BP Inversion soil moisture maps of (c1)–(c3), we can see that the soil moisture at these locations is directly affected by rainfall. The GA-BP inversion soil moisture map of this experiment can better describe the soil moisture change.

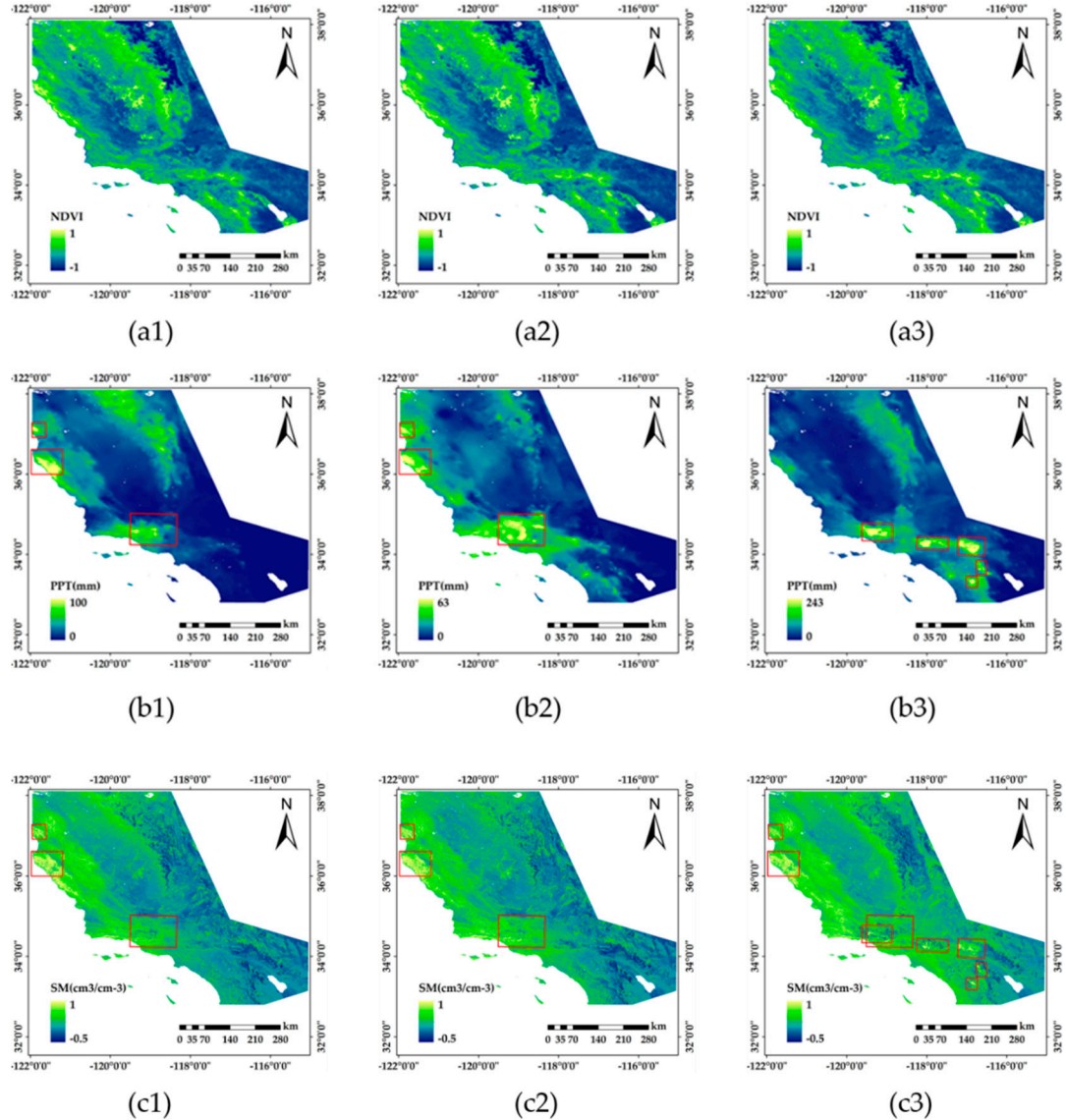

**Figure 8.** Comparison of NDVI, rainfall, and soil moisture in the experimental area during 27 February 2014–1 March 2014; (**a1**–**a3**) are NDVI maps, (**b1**–**b3**) are rainfall distributions during the period, and (**c1**–**c3**) are soil moisture maps obtained from the inversion of this experiment.

### 4.4. Product Accuracy Verification

In order to further evaluate the performance of GA–BP inversion of soil moisture map. We calculate the R-value between the NASA–USDA product and the GA–BP Inversion soil moisture map at 392 fixed points extracted from the study area. Since the NASA–USDA product is only available for three days, for this reason, the GA-BP Inversion soil moisture values were averaged with equal weights. Figure 9 shows the distribution of R-values between NASA–USDA product and the GA–BP Inversion soil moisture at 392 fixed points, and Table 6 shows the statistics of Pearson's coefficient R at 392 sites.

**Table 6.** Pearson's coefficient R statistics for site 392.

| | Accuracy Range | | | | | |
|---|---|---|---|---|---|---|
| R | 0 | 0–0.2 | 0.2–0.4 | 0.4–0.6 | 0.6–0.8 | 0.8–1 |
| Number of sites | 25 | 20 | 22 | 27 | 46 | 253 |

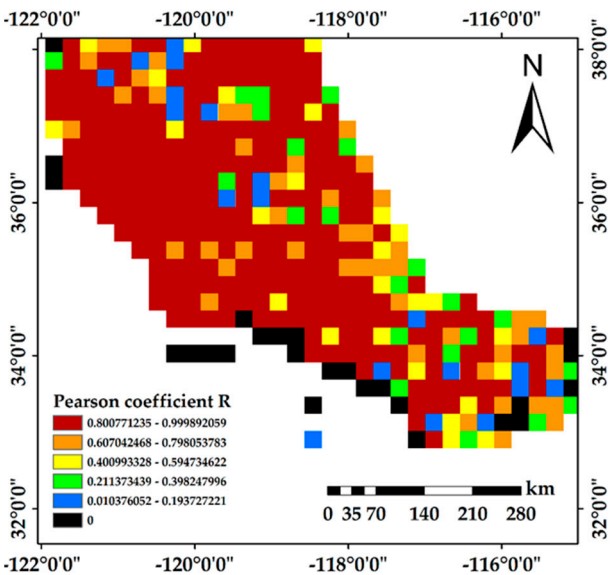

**Figure 9.** Correlation distribution between GA–BP Inversion soil moisture and NASA–USDA products for site 392.

As seen in Table 6, 25 of the 392 fixed sites did not correlate due to the absence of soil moisture fluctuations in NASA-USDA products. A total of 299 of the remaining 367 sites had Pearson correlation coefficients R more significant than 0.6, accounting for 81.5% of the total sites. There were 27 between [0.4–0.6] and 42 less than 0.4. Overall, the multi-data fusion model in this study has strong reliability. As can be seen from Figure 9, the regions with low Pearson coefficients are mainly concentrated in the southeastern part of the experimental area, which is mostly desert and barren areas where the soil moisture inversion is not so ideal. The soil moisture located in the eastern coastal region has a Pearson coefficient greater than 0.8 in most places except for a few areas lacking fluctuations. It can be seen that the GA-BP Inversion soil moisture in this study is consistent with the soil moisture values of NASA products. In conclusion, the multi-data fusion soil moisture inversion model based on GA-BP neural network in this experiment has a particular facilitating effect for predicting soil moisture.

## 5. Conclusions

This study innovatively combines ground-based GRSS-IR soil moisture data with surface environmental data. We construct a multi-data fusion soil moisture model based on the GA-BP neural network to generate a soil moisture product of 500 m per day, which improves existing products' temporal and spatial resolution. Experiments were conducted using data from 1 January 2014–1 March 2014, and 10 geoenvironmental elements (latitude and longitude information, rainfall, air temperature, four topographic factors (elevation, slope, slope direction, and shading), and NDVI) were input into the model. The correlation coefficient R of the optimal model was obtained as 0.8660, and the ubRMSE was 0.0354. The results showed that the GA-BP neural network could better construct the nonlinear relationship between the geoenvironmental elements and soil moisture. Finally, we obtained soil moisture products with a daily spatial resolution of 500 m, compensating for PBO sites' spatial limitation. It also improves the time limit of the existing soil moisture products. We finally obtained maps of the GA-BP inversion soil moisture products with 500 m spatial resolution per day for a total of 15 days during 15 February 2014–1 March 2014. It was analyzed by comparing with the NASA-USDA soil moisture products and rainfall for the same period. The results show that the final generated soil moisture products are more consistent with the NASA-USDA global soil moisture data generated based on microwave remote sensing data. At the same time, the model inversion effect is better in areas with low vegetation cover density and average in areas with high vegetation density or no

vegetation cover. It is also highly consistent with rainfall distribution, and the soil moisture value increases with the increase of rainfall. In conclusion, this paper demonstrates the effectiveness of machine learning methods to obtain high spatial and temporal resolution soil moisture products with great potential in predicting soil moisture through multiple data fusion techniques.

Our work will focus on expanding the test area to cover the whole US or even the entire world in the future. At the same time, we will explore the difference in NDVI calculation in vegetated and non-vegetated areas to further improve the model's accuracy.

**Author Contributions:** Conceptualization, Chao Ren; methodology, Yajie Shi and Chao Ren; software, Yajie Shi; validation, Yajie Shi, Zhiheng Yan and Jianmin Lai; formal analysis, Yajie Shi and Chao Ren; investigation, all authors; re-sources, Chao Ren; data curation, Chao Ren; writing—original draft preparation, Yajie Shi and Chao Ren; writing—review and editing, Yajie Shi and Chao Ren; visualization, Yajie Shi; supervision, Chao Ren; project ad-ministration, Chao Ren; funding acquisition, Chao Ren. All authors have read and agreed to the published version of the manuscript.

**Funding:** This research was funded by the National Natural Science Foundation of China (grant number 42064003).

**Institutional Review Board Statement:** Not applicable.

**Informed Consent Statement:** Informed consent was obtained from all subjects involved in the study.

**Data Availability Statement:** The GNSS-IR soil moisture used in this study are freely available from the International Soil Moisture Network (ISMN). https://ismn.geo.tuwien.ac.at/en/ (accessed on 6 March 2021). All surface environmental data is obtained from Google Earth Engine (GEE). https://code.earthengine.google.com/ (accessed on 9 March 2021).

**Conflicts of Interest:** The authors declare no conflict of interest.

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
