# Peer review of "High Spatial-Temporal Resolution Estimation of Ground-Based Global Navigation Satellite System Interferometric Reflectometry (GNSS-IR) Soil Moisture Using the Genetic Algorithm Back Propagation (GA-BP) Neural Network"

_ijgi, doi:10.3390/ijgi10090623_

Round 1

Reviewer 1 Report

In the manuscript titled: "High Spatial-temporal Resolution Estimation of Ground-based 2 Global Navigation Satellite System Interferometric Reflectome-try(GNSS-IR)Soil Moisture using The Genetic Algorithm Back-Propagation (GABP) Neural Network", the authors present a methodology for soil moisture inversion algorithm based on machine learning. The underlying idea and approach are worthwhile and the considered topic is of current relevance, due to the soil moisture estimation for global water cycle balance. Also, the introduction is well written and the objectives are well characterized.

Some particular comments are given as follows:

Section 2.3 describing the GABP Neural Network is confusing. It must be rewritten or better explained, so that the reader can better understand how the algorithm works. In figure 1, it is not appreciated how GNSS-IR data enters the GABP NN. Regarding this method, did the authors used a specific software package or it was developed by themselves? Please, specify.

Section 3.3 describing “Data pre-processing” is just a summary of preprocessing steps. In this case, some information and algorithms used for deriving GNSS-IR products must be explained and detailed. Information about the reliability of these methods must also be given. is better to contextualize it in its analytical purpose, otherwise, it becomes difficult to understand

Moreover, the reader would expect section 3.3 before 2.3, as the GABP NN integrates all the derived informational layers and in the flow chart of figure 3, this procedure appears after all information has been collected and processed.

Line 221, “also”

The authors mention that the input signals to the GABP NN are latitude, longitude, NDVI, rainfall, air temperature, land cover type, and four topographic factors. However, in figure 3, the reader observes soil moisture from GNSS-IR as a contributing layer to the multifusion algorithm. However, in phase (3) of the preprocessing steps (line 391), soil moisture is specified as an output layer.  So, it must be better clarified how GNSS-IR SM is used in this method. Is it just for training? Please, better explain, as it is confusing.  Similarly, in figure 3, soil moisture from GNSS-IR appears to enter directly the rest of the model, not an output product of the model. Definitely, a better explanation must be given on how SM from GNSS-IR is derived and used.

Regarding the “Results” section, this part is difficult to follow and sometimes unclear. In this sense, it is suggested to revise this part. It is very difficult to understand section 4.2 and the reader gets lost with figure 5. What are fig.5.a#-fig.5.e#? this must be better clarified.

In its present form, some parts of the manuscript lack the necessary clarity that hinders its readability and availability for the users, and some methods and results need to be improved.

Reviewer 2 Report

The manuscript is in rather good shape.

1- Some editorial revisions are necessary. Some of the sentences are incomplete (e.g., Lines 23, 192, 215, 220, 294, etc.)

2- Please define n in Line 214.

Reviewer 3 Report

The manuscript persents an interesting technique for improving soil moisture based on several inputs, and using data merging technique like a neural network. 

The work is fine, I have no concern.  

Round 2

Reviewer 1 Report

Dear authors,

I went through the new version of the manuscript and I have seen that all responses and corrections have been properly addressed. Si, I suggest your manuscript for publication.